# DJ-1 Serves as a Central Regulator of Diabetes Complications

**DOI:** 10.3390/cimb47080613

**Published:** 2025-08-04

**Authors:** Feng Zhou, Jia-Bin Zhou, Tian-Peng Wei, Dan Wu, Ru-Xing Wang

**Affiliations:** Department of Cardiology, The Affiliated Wuxi People’s Hospital of Nanjing Medical University, Wuxi People’s Hospital, Wuxi Medical Center, Nanjing Medical University, Wuxi 214023, China; njmu2024zf@163.com (F.Z.); zhoujiabin2021@163.com (J.-B.Z.); wech612@163.com (T.-P.W.); wudan787999@163.com (D.W.)

**Keywords:** DJ-1, chronic complications, diabetes, therapeutic target

## Abstract

Diabetes mellitus poses a significant global health challenge, primarily due to its chronic metabolic dysregulation, leading to widespread tissue and organ damage. This systemic impact results in a range of complications that markedly reduce patients’ quality of life. Therefore it is critical to understand the mechanisms underlying these complications. DJ-1 (also known as *PARK7*) is a highly conserved multifunctional protein involved in antioxidative defense, metabolic equilibrium, and cellular survival. Recent studies have highlighted that DJ-1 is critically involved in the pathogenesis and progression of diabetic complications, including macrovascular issues like cardiovascular disease and microvascular conditions such as diabetic nephropathy, retinopathy, and neuropathy, suggesting that it may serve as a promising therapeutic target. Importantly, drugs targeting DJ-1 have shown therapeutic effects. This review provides a comprehensive overview of the current under-standing of DJ-1’s role in diabetes-related complications, emphasizing recent research advances.

## 1. Introduction

Diabetes mellitus is a chronic metabolic disorder characterized by persistent hyperglycemia resulting from either absolute or relative insulin deficiency, pancreatic β-cell dysfunction, and insulin resistance [1]. It is recognized as one of the most rapidly escalating public health crises worldwide in the 21st century [2]. Over time, disruptions in glucose and lipid metabolism, coupled with prolonged high blood glucose levels, significantly impair vascular integrity and blood flow. This pathophysiological environment fosters the development of both macrovascular complications (e.g., cardiovascular disease) and microvascular complications (e.g., diabetic nephropathy, retinopathy, and neuropathy), leading to increased rates of blindness, renal failure, and heart disease. These outcomes contribute to a decline in patients’ quality of life and heightened mortality rates [1,2].

DJ-1 (PARK7) is a multifunctional and evolutionarily conserved protein that plays critical roles in oxidative stress response, cellular metabolism, and survival pathways [3]. Dysregulation of DJ-1 has been implicated in various diseases, including neurodegenerative disorders, type 2 diabetes, and certain cancers [4,5,6]. Accumulating evidence supports a significant role for DJ-1 in the onset and progression of diabetic complications [5,7,8], positioning it as a key focus for future research into diabetes pathology. Current drug research on DJ-1 focuses on DJ-1 transfer proteins, DJ-1 binding proteins, and natural or synthetic compounds that enhance DJ-1 expression, offering promising prospects for treating diabetic complications. This review aims to synthesize existing findings regarding DJ-1’s function in diabetic complications, elucidate its biological properties and molecular mechanisms, evaluate current research limitations, and propose future directions to facilitate the development of targeted therapies involving DJ-1.

## 2. Global Trends and Pathogenesis of Diabetes

The global prevalence of diabetes has surged from 4.6% in 2000 to 10.5% in 2021, with an estimated 537 million individuals currently affected [9]. Projections suggest this number will escalate to 783 million by 2045. Additionally, approximately 352 million people exhibit impaired fasting glucose or glucose tolerance, with 5–10% progressing to type 2 diabetes annually. Due to the often subclinical onset of the disease, many individuals remain undiagnosed in the early stages, as they do not exhibit classic symptoms such as polyphagia, polydipsia, polyuria, or unexplained weight loss [10].

Prolonged metabolic imbalance in diabetes contributes to significant damage across various organs and tissues, resulting in both microvascular and macrovascular complications. For instance, diabetic retinopathy is a leading cause of blindness [11], and diabetic nephropathy represents the most common etiology of end-stage kidney disease (ESKD) [12]. Additional complications such as diabetic neuropathy, peripheral arterial disease, and cardiovascular disorders are major contributors to diabetes-associated morbidity and mortality, adversely impacting patient outcomes [10].

The pathophysiological foundation of type 2 diabetes centers on insulin resistance and initial compensatory hyperinsulinemia, followed by the progressive failure of pancreatic β-cells to produce adequate insulin. The interplay between insulin resistance and β-cell dysfunction creates the complex clinical profile of type 2 diabetes [2,13]. Beyond the classical triad of liver, muscle, and pancreatic β-cells, other factors such as altered insulin signaling, increased lipolysis, elevated glucagon levels, enhanced renal glucose reabsorption, and dysregulation of appetite control also contribute significantly to disease pathogenesis [14].

## 3. Structure and Function of DJ-1

DJ-1 is a 19.8 kDa protein belonging to the DJ-1/ThiJ/PfpI family with a high degree of evolutionary conservation [3], together with broad expression across more than 22 human tissues, including the pancreas, kidneys, skeletal muscle, liver, testis, and heart [15]. Initially identified as a suppressor of the oncogene Ras, *PARK7* was later recognized as a causative gene for autosomal recessive early-onset Parkinson’s disease (PD) [16]. DJ-1 is predominantly cytoplasmic but also localizes to mitochondria and, in some cell types, the nucleus [17].

### 3.1. Structure of DJ-1

According to structural data from protein databases (e.g., Q99497 from UniProt database and 2OR3 from Protein data bank), DJ-1 is composed of 11 β-strands (β1-β11) and 8 α-helices (αA-αH), adopting a helix–loop–helix architecture and functioning primarily as a homodimer [16,18] (Figure 1A). The protein’s core consists of six parallel β-strands flanked by peripheral α-helices. αA is situated centrally within the dimer, while αH is positioned near the carboxyl terminus. A hairpin structure formed by β3 and β4 facilitates dimerization [15,19] (Figure 1B). The terminal three amino acids, particularly residues L187 and K188, play a crucial role in stabilizing the DJ-1 dimer [16]. A point mutation at residue L166 (L166P) within the αG helix disrupts dimerization and destabilizes the protein, accelerating its degradation and contributing to early-onset familial Parkinson’s disease [15]. While other missense mutations may affect DJ-1 stability, they generally do not appear to disrupt the protein’s dimeric form to the same extent [20]. Emerging evidence suggests that the α-helices modulate the accessibility of DJ-1’s putative catalytic sites, thereby influencing its enzymatic function. Under oxidative stress, DJ-1 undergoes conformational changes that alter its catalytic activity, enabling it to neutralize reactive oxygen species. This oxidative responsiveness may underlie DJ-1’s cytoprotective functions [20].

### 3.2. Functions of DJ-1

DJ-1 is a multifunctional protein predominantly localized in the cytoplasm, although it is also present in mitochondria. It primarily acts as a cysteine-dependent protease and participates in diverse cellular roles, ranging from redox regulation to transcriptional coactivation. Notably, in certain cancers, DJ-1 undergoes functional transformation into a decarboxylase enzyme, highlighting its context-dependent activity [16,17,21]. Functionally, DJ-1 is implicated in various physiological and pathological processes, including apoptosis regulation, transcriptional modulation, redox homeostasis, and cell proliferation [22].

Substantial evidence indicates that DJ-1 plays a pivotal role in cellular defense against oxidative stress. One key mechanism involves its translocation to mitochondria in response to oxidative stress, a process mediated by its three redox-sensitive cysteine residues, including Cys46, Cys53, and particularly Cys106. These residues undergo reversible oxidation to sulfinic (-SO_2_H) or sulfonic (-SO_3_H) acid forms, thereby facilitating the direct scavenging of reactive oxygen species (ROS) [5,19,23]. In parallel, DJ-1 contributes to antioxidant defense by modulating the Nrf2 (nuclear factor erythroid 2-related factor 2) signaling pathway. DJ-1 promotes the dissociation of Nrf2 from its cytoplasmic inhibitor Keap1 (Kelch-like ECH-associated protein 1), allowing Nrf2 to translocate to the nucleus. There, it binds to antioxidant response elements (AREs) to activate the transcription of various cytoprotective genes, including heme oxygenase-1 (*HO-1*), NAD(P)H quinone dehydrogenase 1 (*NQO1*), and the catalytic subunit of glutamate–cysteine ligase (*GCLC*) [8,19,24].

Among DJ-1’s redox-sensitive residues, Cys106 plays a particularly crucial role. Under conditions of moderate oxidative stress, Cys106 is oxidized to a stable sulfinic acid form (Cys106-SO_2_H), which facilitates DJ-1’s mitochondrial localization and mitigates oxidative damage by modulating apoptosis signal-regulating kinase 1 (ASK1) activity [15,25]. However, hyperoxidation to the sulfonic acid form (Cys106-SO_3_H) induces a conformational shift that abrogates DJ-1’s enzymatic and redox sensor functions, effectively inactivating the protein [12,26].

In addition to its redox regulatory roles, DJ-1 has been shown in vitro to possess glyoxalase-like enzymatic activity. It can detoxify harmful aldehydes such as glyoxal (GO) and methylglyoxal (MGO), which are byproducts of lipid peroxidation and glycolysis, by converting them into less toxic compounds like glycolic acid and lactic acid, respectively. This detoxification mechanism serves to mitigate glycation damage to proteins and DNA [5,16,21,27]. Cells deficient in DJ-1 display elevated levels of DNA glycation, increased DNA strand breaks, and heightened p53 phosphorylation, indicating increased genotoxic stress [12]. DJ-1 is also involved in cell survival signaling. It has been shown to activate the ERK1/2 and PI3K/Akt pathways following oxidative stimuli. Upon activation, DJ-1 facilitates the nuclear translocation of ERK1/2, leading to phosphorylation of transcription factor Elk1 and upregulation of antioxidant enzyme superoxide dismutase 1 (SOD1). These effects collectively bolster cellular antioxidant defenses. Additionally, DJ-1 suppresses apoptosis by inhibiting ASK1- and p53-mediated signaling cascades [28]. Other studies suggest that DJ-1 contributes to cellular homeostasis by repressing p53 expression, preserving mitochondrial integrity, alleviating endoplasmic reticulum (ER) stress, and modulating protein turnover through interactions with the ubiquitin–proteasome system [29].

## 4. DJ-1 and the Pathological Mechanisms of Diabetic Complications

While ROS at physiological levels play essential roles in glucose metabolism and signaling, persistent elevation of ROS due to chronic hyperglycemia leads to sustained oxidative stress, contributing to the pathogenesis of diabetes and its complications. Pancreatic β-cells, characterized by inherently low antioxidant capacity, are particularly vulnerable to oxidative insults [30]. In this context, DJ-1 serves a protective role by safeguarding β-cells from various forms of cellular stress, including oxidative, ER stress, and cytotoxic insults from streptozotocin and proinflammatory cytokines, thereby preserving β-cell viability and insulin secretory function [31].

Type 2 Diabetes Mellitus (T2DM) significantly increases the risk of both microvascular (e.g., retinopathy, neuropathy, and nephropathy) and macrovascular (e.g., coronary artery disease, cerebrovascular events, and peripheral arterial disease) complications [2]. Although the precise molecular pathways through which hyperglycemia damages the vasculature remain incompletely defined, current evidence implicates oxidative stress as a central mediator. Chronic high-glucose conditions enhance ROS production, perpetuating a deleterious cycle that disrupts multiple signaling pathways, including activation of the polyol pathway, formation of advanced glycation end products (AGEs), activation of protein kinase C, and increased hexosamine pathway flux [1,32]. DJ-1 exerts complex, dualistic effects in diabetic complications. On one hand, it has protective effects, enhancing energy metabolism and delaying the onset of diabetic complications. On the other, some studies suggest that DJ-1 may contribute to fibrosis and exacerbate disease progression under certain conditions. The following sections summarize the molecular mechanisms through which DJ-1 influences diabetic complications, which are also presented in Table 1.

### 4.1. Diabetic Retinopathy

Diabetic retinopathy (DR) is a prevalent microvascular complication affecting approximately 35% of individuals with diabetes and stands as a leading cause of blindness in older populations worldwide [1,10,38]. Chronic hyperglycemia damages retinal vasculature, leading to microaneurysms, hemorrhages, retinal detachment, and ultimately vision loss. DR progresses from the early non-proliferative stage, marked by capillary leakage and microvascular weakening, to the more advanced proliferative stage characterized by aberrant neovascularization in the retina and vitreous body [38].

Reduced DJ-1 expression has been implicated in DR pathogenesis. In retinal pericytes (RPs), hyperglycemic conditions promote oxidative stress, mitochondrial depolarization, and increased membrane permeability, resulting in cytochrome c and BAX release from mitochondria, triggering both caspase-dependent and -independent apoptosis pathways [8]. Wang et al. demonstrated that DJ-1 mitigates high-glucose-induced oxidative damage and RP apoptosis by activating the Nrf2 pathway [8]. This protective effect may be mediated by DJ-1’s interaction with nuclear- and mitochondria-encoded subunits of complex I, helping maintain its structural and functional integrity and preventing the release of pro-apoptotic factors [39]. Zeng and colleagues reported that DJ-1 overexpression activates the PI3K/AKT/mTOR pathway, enhances MnSOD and catalase (CAT) expression and activity, suppresses ROS production, and downregulates apoptotic markers such as phosphorylated p53 and activated caspase-3, thereby protecting RPs from high-glucose-induced apoptosis [29].

Beyond its effects on RPs, DJ-1 also protects other retinal components. For example, DJ-1 deficiency has been shown to worsen retinal ganglion cell (RGC) loss in type 1 diabetic models, while its overexpression reduces RGC apoptosis and preserves neuroretinal integrity, underscoring its role in mitochondrial maintenance in RGCs [7]. Increased permeability of retinal capillary endothelial cells, which is another hallmark of DR [38], is exacerbated in T2DM. Here, decreased DJ-1/Nrf2/NQO1 expression and reduced Nrf2 nuclear translocation contribute to heightened caspase-3 activity and p53 phosphorylation in T2DM, increasing susceptibility to oxidative injury in retinal and corneal endothelial cells [32,40]. More recently, Wu et al. engineered a novel nanocarrier system by conjugating tetrahedral framework nucleic acids (tFNAs) with small activating RNAs targeting DJ-1 (DJ-1 saRNAs). This composite, termed tFNAs-DJ-1 saRNAs, was shown in vitro to enhance DJ-1 expression in retinal pigment epithelium (RPE) cells. Following oxidative stress induced by hydrogen peroxide (H_2_O_2_), transfection with tFNAs-DJ-1 saRNAs decreased ROS accumulation, improved cell viability, and reduced apoptosis. Additionally, this intervention preserved mitochondrial structure and function under stress, potentially through the activation of the Erk and Nrf2 pathways [17].

### 4.2. Diabetic Nephropathy

Diabetic nephropathy (DN), the most prevalent microvascular complication associated with advanced diabetes, is also a leading contributor to end-stage renal disease. Histologically, DN is characterized by glomerular hypertrophy, thickening of the glomerular basement membrane, mesangial expansion, glomerulosclerosis in the terminal stage, and interstitial fibrosis [1,38,41]. Its pathogenesis is multifactorial, involving dysregulated glucose metabolism, oxidative stress, hemodynamic disturbances, genetic predispositions, inflammatory mediators, and the accumulation of AGEs [42].

An increasing body of evidence suggests that DJ-1 plays a significant role in the onset and progression of DN [41,43,44]. In a streptozotocin (STZ)-induced diabetic rat model, Sun et al. observed that DJ-1 protein expression in renal tissues underwent dynamic changes throughout disease progression. During the early stages of DN, DJ-1 levels were upregulated as a compensatory antioxidant response. This increase facilitates the dissociation of Nrf2 from its cytoplasmic inhibitor Keap1, enabling Nrf2 nuclear translocation and binding to AREs. This interaction promotes the transcription of stress-response and antioxidant genes, thereby exerting a protective effect against oxidative stress. However, in the later stages of DN, DJ-1 expression may decline, resulting in exacerbated cellular oxidative damage and diminished defense capacity in T1DM [34]. Conversely, experimental overexpression of DJ-1 has been shown to enhance Nrf2 and HO-1 expression, thereby protecting renal tubular epithelial cells from hyperglycemia-induced oxidative injury [45].

Intriguingly, DJ-1 has also been implicated in renal fibrosis, potentially acting downstream of transforming growth factor-beta (TGF-β). Das et al. demonstrated that DJ-1 is a transcriptional target of the Smad3 signaling cascade activated by TGF-β in diabetic kidneys. Through upregulation of DJ-1, TGF-β enhances protein kinase C-βII (PKCβII) phosphorylation, which in turn activates mTORC2. This activation facilitates Akt phosphorylation at its hydrophobic motif, driving the expression of collagen I, stimulating proximal tubular epithelial cell hypertrophy, and promoting extracellular matrix accumulation, hallmarks of renal fibrosis [12]. Further investigations revealed that under hyperglycemic conditions, DJ-1 interacts with PTEN at the PDGFRβ docking site, triggering the PI3K/Akt/mTORC1 signaling pathway. This cascade promotes hypertrophic responses and the expression of fibrotic markers such as fibronectin and collagen I. Silencing DJ-1 was found to inhibit PDGFRβ tyrosine phosphorylation and downstream PI3K activation, effectively mitigating these pathogenic effects [41].

### 4.3. Diabetic Neuropathy

Diabetic neuropathy (DN), a common and insidious chronic complication of diabetes, can affect virtually any component of the nervous system. It encompasses a wide array of clinical syndromes, including diabetic peripheral neuropathy (DPN), autonomic neuropathy, small fiber neuropathy, and multiple radiculopathies [38]. As with diabetic cardiomyopathy and DN, intricate molecular interactions are central to its pathogenesis. A pivotal factor is the accumulation of reactive α-dicarbonyl compounds (α-DCs), such as GO, MGO, and AGEs, which are widely recognized as key mediators of diabetic complications [46]. Chaudhuri et al. identified the transient receptor potential ankyrin 1 (TRPA1) as a sensor for α-DCs. Upon activation, TRPA1 triggers the calcium-dependent kinase signaling cascade that activates the transcription factor SKN-1, which in turn upregulates DJ-1 expression. As a glyoxalase-like enzyme, DJ-1 reduces α-DC-induced toxicity, thereby mitigating neuronal damage [47]. Mitochondrial dysfunction and oxidative stress are also central mechanisms in diabetic neuropathy. Natural compounds such as polydatin, rutin, and resveratrol have demonstrated protective effects by stimulating the Nrf2 signaling pathway [48]. DJ-1, situated upstream of Nrf2, regulates oxidative stress responses and may contribute to neuroprotection in diabetic settings by modulating Nrf2 activity and functioning as a glyoxalase enzyme.

### 4.4. Macrovascular Complications of Diabetes

Individuals with diabetes face a significantly (2- to 10-fold) heightened risk of experiencing cardiovascular events, including coronary heart disease (CHD), myocardial infarction, heart failure, and stroke, compared to the general population [1]. Moreover, the mortality rate from cardiovascular causes is approximately 1.13 times higher in diabetic patients than in their non-diabetic counterparts [10]. These macrovascular complications, driven by chronic hyperglycemia, predominantly affect large vessels such as coronary and cerebral arteries, and are major contributors to diabetes-related mortality [1].

As early as 2009, Collins et al. demonstrated that aging exacerbates high-fat diet (HFD)-induced vascular dysfunction, and that DJ-1 deficiency enhances atherosclerosis susceptibility [33]. More recently, Sun et al. reported that the diabetic milieu upregulates N-acetyltransferase 10 (NAT10), which facilitates acetylation of DJ-1 at lysine 93. This post-translational modification promotes DJ-1 ubiquitination at lysine 41 via acetylation–ubiquitination crosstalk, resulting in DJ-1 degradation. Reduced DJ-1 levels subsequently activate the DJ-1/STAT1/STAT2 microcalcification axis. Specifically, STAT1 becomes hyperphosphorylated and translocates into the nucleus, where it enhances transcription of the *Ager* gene encoding the receptor for AGEs (RAGE). Upregulated RAGE expression promotes the release of matrix vesicles (MVs) enriched in milk fat globule-EGF factor 8 protein (MFGE8), which mediates collagen–MV interactions and drives mineralization and microcalcification within diabetic plaques in T2DM [5]. Of note, overexpression of DJ-1 inhibits VSMC calcification under hyperglycemic conditions but has a negligible impact under normoglycemic conditions, suggesting a context-specific protective role confirmed in this prior study. The influence of DJ-1 on VSMC phenotype was further corroborated by a study demonstrating that VSMCs from DJ-1-deficient ApoE-/- mice exhibit a marked reduction in the contractile phenotype and an increase in the synthetic phenotype, underscoring DJ-1’s pivotal role in VSMC phenotypic modulation [49].

Myocardial ischemia–reperfusion injury (IRI) remains a pressing global health concern, as the restoration of blood supply following ischemic episodes can paradoxically worsen myocardial damage. Ischemic postconditioning (IPostC)—a sequence of brief, intermittent cycles of ischemia and reperfusion applied at the onset of reperfusion—has demonstrated myocardial protective effects in non-diabetic animal models, primarily through the attenuation of oxidative stress. However, in diabetic settings, this protective mechanism is notably compromised, largely due to the impaired activation of antioxidant defense pathways in T1DM [50]. A pivotal study by Liu et al. was the first to identify that hyperglycemia-mediated suppression of DJ-1 expression disrupts the endogenous cardiac protective mechanisms in diabetic rats subjected to ischemia [37]. Expanding on these findings, Li et al. observed that diabetic rats exhibited significantly greater infarct sizes and elevated oxidative damage following I/R injury as compared to non-diabetic counterparts. These detrimental outcomes were associated with a marked downregulation of cardiac DJ-1 and an upregulation of PTEN. Remarkably, cardiac-specific overexpression of DJ-1 via AAV9 reinstated the cardioprotective efficacy of IPostC. This was accompanied by enhanced translocation of DJ-1 from the cytoplasm to both the nucleus and mitochondria, decreased PTEN expression, and activation of the Nrf2/HO-1 transcriptional axis. In contrast, targeted deletion of DJ-1 using AAV9 vectors exacerbated myocardial I/R injury in diabetic hearts. Intriguingly, pharmacological inhibition of PTEN or activation of Nrf2 partially mitigated this exacerbation, suggesting that DJ-1 confers protection by modulating these downstream signaling molecules. Furthermore, DJ-1 was shown to promote its own translocation to subcellular organelles in diabetic myocardium by suppressing PTEN, thereby rescuing the cardioprotective effect of IPostC. Notably, this protective mechanism was abrogated when DJ-1 carried the C106S point mutation, highlighting the critical importance of the C106 residue in preserving its redox-sensitive functionality [36] in T1DM. Supporting these observations, Li et al. demonstrated that N-acetylcysteine (NAC), a precursor of the endogenous antioxidant glutathione (GSH), alleviated myocardial I/R injury in diabetic models by upregulating DJ-1 expression and enhancing activation of the PTEN/Akt signaling cascade [35] in T1DM. Collectively, these studies suggest that the attenuation of IPostC-induced cardioprotection in diabetes may be attributed to reduced DJ-1 expression and impaired oxidative stress response. The integrity of the C106 site on DJ-1 appears essential for its cardioprotective effects in diabetic myocardial tissue [51].

## 5. Clinical Prospects and Challenges

Current research on DJ-1 drug targets focuses on DJ-1 delivery proteins, DJ-1 binding proteins, and chemical agents that enhance DJ-1 expression. For instance, tFNAs-DJ-1 saRNAs have been shown to increase DJ-1 expression in retinal pigment epithelial cells, thereby reducing reactive oxygen species and apoptosis. Moreover, maintaining DJ-1’s Cys106 in the Cys106-SO2H oxidation state, rather than Cys106-SO3H, preserves its protective function against oxidative stress. Antioxidants such as vitamins E and C, coenzyme Q10, and resveratrol help mitigate oxidative stress [52]. ND-13, a DJ-1-derived peptide, has shown promise in treating Parkinson’s disease by activating the Nrf2 antioxidant pathway, lowering reactive oxygen species, and enhancing motor function [53]. Additionally, ND-13 aids in recovery from focal ischemia [52]. NAC has been reported to elevate DJ-1 levels, inhibit PTEN expression, activate the PI3K/AKT pathway, and improve diabetic myocardial ischemia–reperfusion injury [35].

Nevertheless, several important limitations remain in the current body of research. First, the regulatory mechanisms controlling DJ-1 expression across different tissues have not been fully elucidated. Additionally, there is a paucity of sequencing analyses investigating specific pathways, and current research evidence is insufficient, necessitating further exploration. Second, the majority of existing evidence is derived from in vitro or preclinical animal studies, with a lack of robust clinical cohort data to substantiate its relevance in human diabetic pathology. Another critical area that remains unexplored is the potential role of DJ-1 in metabolic memory, which is an enduring epigenetic phenomenon that perpetuates diabetic complications despite glycemic control. Whether DJ-1 participates in epigenetic regulation through mechanisms such as DNA methylation or histone modification is still unknown. Furthermore, since diabetes frequently coexists with neurodegenerative diseases such as Alzheimer’s disease, investigating the shared molecular pathways involving DJ-1 in these comorbidities could yield significant clinical insights. Future investigations into the interactions between DJ-1 and cellular pathways are essential. In vivo studies are needed to confirm the therapeutic potential of targeting DJ-1 in preclinical models and clinical applications.

## 6. Conclusions

In recent years, DJ-1 has gained increasing attention for its multifaceted role as an antioxidant and molecular chaperone, particularly in the context of diabetes-related chronic complications. Accumulating evidence indicates that DJ-1 contributes to the protection of various organs against diabetic damage, including the kidneys, retina, peripheral nerves, and large blood vessels. These protective effects are primarily mediated through the modulation of oxidative stress, inflammation, mitochondrial dynamics, and apoptotic signaling (Figure 2). These findings underscore the potential of DJ-1 as an intrinsic cytoprotective factor in the pathogenesis of diabetes-related complications. As interdisciplinary research advances, DJ-1 holds promise as a novel therapeutic target for the precision prevention and treatment of chronic complications arising from diabetes.

## Figures and Tables

**Figure 1 cimb-47-00613-f001:**
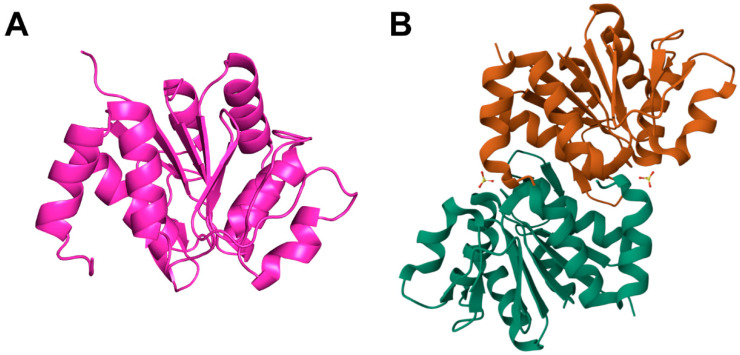
**The crystal structure of DJ-1.** (**A**) Spatial structure of the DJ-1 protein from UniProt database (ID: Q99497). (**B**) The dimerization arrangement of DJ-1 (PDB: 2OR3). All the data above are from homo sapiens.

**Figure 2 cimb-47-00613-f002:**
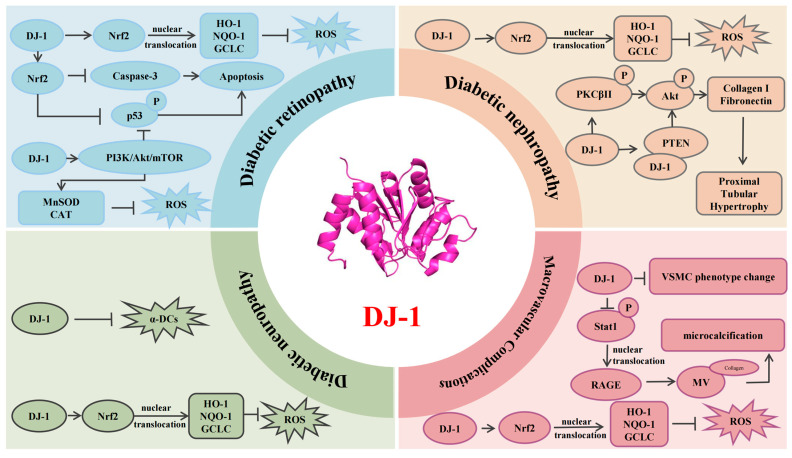
**DJ-1’s role in diabetic complications.** DJ-1, a multifunctional intracellular protein, plays a critical role in protective responses during diabetic complications. It facilitates the release of Nrf2 from the cytoplasmic inhibitor Keap1, allowing Nrf2 to move to the nucleus and bind to antioxidant response elements. This activation leads to the transcription of cytoprotective genes (*HO-1*/*NQO-1*/*GCLC*), reducing ROS production and oxidative stress. Furthermore, Nrf2 inhibits apoptosis by suppressing p53 phosphorylation and caspase-3 activation. DJ-1 also curtails ROS production by activating the PI3K/Akt/mTOR pathway, enhancing MnSOD and CAT expression and activity. In diabetic nephropathy, DJ-1 promotes Akt phosphorylation by aiding PKCβII phosphorylation and its interaction with PTEN, which induces collagen I and fibronectin expression, causing proximal tubular hypertrophy. In diabetic peripheral neuropathy, DJ-1 functions as a glyoxalase-like enzyme, mitigating α-DC-induced toxicity and reducing neuronal damage. In diabetic macrovascular complications, DJ-1 prevents vascular smooth muscle cells (VSMCs) from shifting from a contractile to a synthetic phenotype and inhibits Stat1 phosphorylation and nuclear translocation, thereby reducing RAGE expression. Lower RAGE levels decrease microvesicle (MV) release, reducing vascular plaque calcification in diabetes. Blunt-ended lines indicate inhibitory effects, while arrows represent stimulatory effects. Nrf2, nuclear factor erythroid 2-related factor 2; HO-1, heme oxygenase-1; NQO-1, NAD(P)H quinone dehydrogenase 1; GCLC, glutamate–cysteine ligase; PI3K, phosphoinositide 3-kinase; Akt, Ak strain transforming; mTOR, mechanistic target of rapamycin; MnSOD, manganese superoxide dismutase; ROS, reactive oxygen species; CAT, catalase; VSMC, vascular smooth muscle cell; PKCβII, protein kinase C beta II; PTEN, phosphatase and tensin homolog; α-DCs, α-dicarbonyl compounds; Stat1, signal transducer and activator of transcription 1; RAGE, receptor for advanced glycation end products; MVs, matrix vesicles.

**Table 1 cimb-47-00613-t001:** Interactions between DJ-1 and other proteins.

DJ-1 Protein Partners	Experimental Species	Biological Effect	References
Nuclear factor erythropoiesis-related factor 2 (NRF2)	*Mus musculus* *Rattus norvegicus*	DJ-1 facilitates NRF2 release from KEAP1, promoting its nuclear translocation. NRF2 then binds AREs, enhancing transcription of oxidative stress-responsive genes such as *HO-1* and *NQO1*, thereby mitigating hyperglycemia-induced oxidative stress.	[8,33,34]
	*Mus musculus*	NRF2 translocation reduces caspase-3 activity and p53 phosphorylation, decreasing the susceptibility of corneal endothelial cells to UVA-induced oxidative injury.	[32]
Phosphatase and tensin homolog (PTEN)	*Rattus norvegicus*	NAC alleviates diabetic myocardial I/R injury via DJ-1-mediated activation of the PTEN/Akt pathway.	[35]
	*Rattus norvegicus*	DJ-1 overexpression re-establishes IPostC-induced cardioprotection in diabetes through mitochondrial and nuclear translocation, suppressing PTEN and influencing cell survival and immune regulation.	[36]
Phosphatidylinositol 3-kinase (PI3K)	*Rattus norvegicus*	DJ-1 may exert cytoprotective effects by activating the PI3K/Akt pathway, which governs key cellular functions such as migration, metabolism, and survival.	[37]
Signal transducer and activator of transcription 1 (STAT1)	*Mus musculus* *Homo sapiens*	The interplay between NAT10-mediated acetylation and TRIM32-mediated ubiquitination drives the proteasomal degradation of DJ-1. This activates the DJ-1/STAT1/RAGE axis, promoting coronary microcalcification via osteogenic transdifferentiation.	[5]

## Data Availability

No datasets were generated or analyzed during the current study.

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
