# Peer review of "DJ-1 Serves as a Central Regulator of Diabetes Complications"

_cimb, 2025, doi:10.3390/cimb47080613_

Round 1
Reviewer 1 Report
Comments and Suggestions for Authors
The authors highlight the role of DJ1 in progression of complications of diabetes mellitus, as it is involved in oxidative stress response, cellular metabolism and survival pathways. The goal of the review was to summarize the role of DJ1 in diabetic complications, analyze research limitations and propose future research directions for targeted therapy. The authors highlight in depth the role of DJ1 in preventing various microvascular complications such as retinopathy, nephropathy and neuropathy as well as macrovascular complications such as coronary heart disease
Overall this was a very helpful review in understanding the role of DJ-1 in various complications of diabetes mellitus, especially that the authors made the distinction between microvascular and macrovascular complications. They also very effectively highlighted several limitations in current research such as the lack of human cohort data and the unknown role of DJ1 in metabolic memory. Further areas of improvement include:
1. The authors often do not clarify when they are discussing type 1 vs type 2 diabetes. Despite implying at times that DJ1 is implicated in both type 1 and type 2 diabetes, most of the research appears to be in type 2 diabetes.
2. It would also be helpful to have a diagram of DJ1’s structure as the authors go in depth in describing its structure.
3. Overall the authors did well in highlighting limitations in research, however, they state that downstream effectors and direct molecular targets are unknown despite describing various mechanisms involving DJ1. It would be helpful to provide examples of possible downstream effectors and direct molecular targets to explore
4. Despite stating that proposing future research directions is their goal, the authors do not go into detail of possible areas where further research can be done, instead stating that DJ-1 holds promise as a possible target for prevention and treatment of diabetic complications. It would be helpful for the authors to go into more detail about possible future directions in DJ-1 research.
Author Response
Dear Reviewer:
Thank you for your letter and for your comments concerning our manuscript entitled "DJ-1 serves as a central regulator of diabetes complications". Those comments are all valuable and very helpful for revising and improving our paper, as well as the important guiding significance to our research. We have studied comments carefully and have made correction which we hope to meet with approval. Revised portions are marked in red in the paper. The main corrections in the paper and the responses to the reviewer’s comments are as following:
1. The authors often do not clarify when they are discussing type 1 vs type 2 diabetes. Despite implying at times that DJ-1 is implicated in both type 1 and type 2 diabetes, most of the research appears to be in type 2 diabetes.
Answer: We sincerely appreciate your recognition of our research. As you indicated, DJ-1 has been implicated in both T1DM and T2DM. Therefore, in this manuscript we have implemented distinct labeling of T1DM and T2DM for all instances referenced throughout the literature review. Revision has been made according to the comment and all the changes were marked in red color in the Marked-up Revised Manuscript.
2. It would also be helpful to have a diagram of DJ-1’s structure as the authors go in depth in describing its structure.
Answer: Special thanks to you for your good comments. It is indeed true that it would be helpful to have a diagram of DJ-1’s structure as the authors go in depth in describing its structure. We have constructed a schematic diagram of DJ-1’s structural architecture to provide readers with enhanced clarity regarding its molecular composition.​ Revision has been made according to the comment and all the changes were marked in red color in the Marked-up Revised Manuscript (Page 15, figure 1).
3. Overall the authors did well in highlighting limitations in research, however, they state that downstream effectors and direct molecular targets are unknown despite describing various mechanisms involving DJ-1. It would be helpful to provide examples of possible downstream effectors and direct molecular targets to explore.
Answer: Thanks for your careful review and pretty good suggestion. As you suggested, we conducted targeted investigations into potential downstream effectors and direct molecular targets, accompanied by detailed schematic representations. Revision has been made according to the comment and all the changes were marked in red color in the Marked-up Revised Manuscript (Page 15-16, figure 2).
4. Despite stating that proposing future research directions is their goal, the authors do not go into detail of possible areas where further research can be done, instead stating that DJ-1 holds promise as a possible target for prevention and treatment of diabetic complications. It would be helpful for the authors to go into more detail about possible future directions in DJ-1 research.
Answer: We really appreciate for your warm work earnestly. It is really true as you suggested that more details about possible future directions in DJ-1 research will be helpful for the authors. Therefore, we have established a dedicated section in the manuscript to consolidate current clinical applications of DJ-1, while delineating future prospects and persisting challenges—thus highlighting its significance as a putative therapeutic target for preventive and therapeutic management of diabetic complications​ (Page 11-12, line 306-331).
Finally, it is a great honor for us to have met such a conscientious and responsible reviewer as you. We have tried our best to improve the manuscript and made changes in the manuscript. Meanwhile, we appreciate for your warm work earnestly, and hope that the correction will meet with approval.
Once again, thank you very much for your comments and suggestions.
Best wishes to you!
Reviewer 2 Report
Comments and Suggestions for Authors
The author reviewed the mechanism of action of DJ-1 in regulating diabetic complications. However, the current literature review has not yet met the basic requirements for academic paper publication. Specific suggestions are as follows:
- Uncommon abbreviations should not be used in the title. When an abbreviation appears for the first time, its full form should be provided.
- Judging from the content, this manuscript should be a review rather than an Article.
- The Introduction section by the author is overly brief. The author should analyze the main drug targets and their applications in current diabetes treatments to introduce the significant role of DJ-1 in improving diabetic complications.
- The author should specify in which complications DJ-1 plays a crucial role and mention this in the abstract.
- In the sections on structure and mechanism of action, the author should draw mechanism diagrams to clarify the mechanism of action and potential signaling pathways of DJ-1.
- When conducting a literature review on the mechanism of action of DJ-1 in different complications, the author should also draw mechanism diagrams to illustrate the mechanism of action of DJ-1.
- Additionally, the most crucial aspect is the feasibility of developing drugs with DJ-1 as a therapeutic target. The author should analyze the current research progress on potential functional compounds and other related areas.
Author Response
Dear Reviewer:
Thank you for your letter and for your comments concerning our manuscript entitled "DJ-1 serves as a central regulator of diabetes complications". Those comments are all valuable and very helpful for revising and improving our paper, as well as the important guiding significance to our research. We have studied comments carefully and have made correction which we hope to meet with approval. Revised portions are marked in red in the paper. The main corrections in the paper and the responses to the reviewer’s comments are as following:
1. Uncommon abbreviations should not be used in the title. When an abbreviation appears for the first time, its full form should be provided.
Answer: We sincerely appreciate your recognition of our research. We have carefully reviewed our manuscript to ensure that all abbreviations are defined upon their first occurrence in each of the following sections: the abstract, the main text, all figures, and tables. Notably, DJ-1 is the official gene symbol designated by the HUGO Gene Nomenclature Committee (HGNC) and the standard nomenclature in Parkinson's disease research. Its usage without expansion is universally accepted in peer-reviewed literature. For clarity, we have added in parentheses: DJ-1 (PARK7) in its first mention, as 'PARK7' is its alias in the context of Parkinson's disease-associated genes. However, the symbol 'DJ-1' remains the primary term. Revision has been made according to the comment and all the changes were marked in red color in the Marked-up Revised Manuscript.
2. Judging from the content, this manuscript should be a review rather than an Article.
Answer: We sincerely appreciate the reviewer's thorough evaluation of our manuscript. We wish to clarify that this submission was ​​explicitly categorized as a review​​ during the initial submission process, in full accordance with the journal's guidelines for this article type. Our work synthesizes existing evidence on DJ-1's role in diabetes complications, aligning with the core purpose of a review.
3. The Introduction section by the author is overly brief. The author should analyze the main drug targets and their applications in current diabetes treatments to introduce the significant role of DJ-1 in improving diabetic complications.
Answer: Thanks for your careful review and pretty good suggestion. We have enhanced the introduction by incorporating the primary drug targets of DJ-1 in current diabetes treatments and their applications, thereby underscoring DJ-1's critical role in ameliorating diabetic complications. Revision has been made according to the comment and all the changes were marked in red color in the Marked-up Revised Manuscript (Page 3, line 63-65).
4. The author should specify in which complications DJ-1 plays a crucial role and mention this in the abstract.
Answer: We really appreciate for your warm work earnestly. As you suggested, we have added content in the abstract section about in which diabetic complications DJ - 1 plays a key role to help readers better understand the content. Revision has been made according to the comment and all the changes were marked in red color in the Marked-up Revised Manuscript (Page 2, line 31-34).
5. In the sections on structure and mechanism of action, the author should draw mechanism diagrams to clarify the mechanism of action and potential signaling pathways of DJ-1.
Answer: Special thanks to you. We have included separate mechanism diagrams for the structural (Page 15, figure 1) and action aspects (Page 15-16, figure 2) of DJ-1 in the manuscript to clarify its mechanism of action and potential signaling pathways.
6. When conducting a literature review on the mechanism of action of DJ-1 in different complications, the author should also draw mechanism diagrams to illustrate the mechanism of action of DJ-1.
Answer: We really appreciate for your warm work earnestly. In the manuscript, we have analyzed and organized the mechanisms of DJ-1 across various diabetic complications, creating a diagram to elucidate its mode of action (Page 15-16, figure 2).
7. Additionally, the most crucial aspect is the feasibility of developing drugs with DJ-1 as a therapeutic target. The author should analyze the current research progress on potential functional compounds and other related areas.
Answer: Thanks to you for reminder. It is really true that the most crucial aspect is the feasibility of developing drugs with DJ-1 as a therapeutic target. In our paper, we have explored the feasibility of targeting DJ-1 with drugs. Additionally, we have analyzed current research on related fields, including potential functional compounds, and discussed future application challenges. This aims to underscore the importance of DJ-1 as a target for preventing and treating diabetic complications. Revision has been made according to the comment and all the changes were marked in red color in the Marked-up Revised Manuscript (Page 11-12, line 306-331).
​In response to reviewers’ editorial concerns regarding linguistic quality, we have engaged professional native-English editing services to enhance manuscript rigor. The corresponding certification has been deposited in the attachment.​
Finally, it is a great honor for us to have met such a conscientious and responsible reviewer as you. We have tried our best to improve the manuscript and made changes in the manuscript. Meanwhile, we appreciate for your warm work earnestly, and hope that the correction will meet with approval.
Once again, thank you very much for your comments and suggestions.
Best wishes to you!

Round 2
Reviewer 2 Report
Comments and Suggestions for Authors
I appreciate the author for adopting my suggestions. I believe the author has made relatively good revisions based on my recommendations. Building on this, I think the mechanism diagram in Figure 2 still needs further refinement to distinguish whether the relevant action mechanisms involve transmembrane processes and whether they pertain to regulatory events within the cell nucleus. Additionally, the author should clearly differentiate and explain the upstream-downstream relationships of the signaling pathways in the Figure 2. After the author improves the mechanism diagram, I believe the manuscript could be considered for acceptance.
Author Response
Dear Reviewer:
Thank you for your letter and for your comments concerning our manuscript entitled "DJ-1 serves as a central regulator of diabetes complications". Those comments are all valuable and very helpful for revising and improving our paper, as well as the important guiding significance to our research. We have studied comments carefully and have made correction which we hope to meet with approval. Revised portions are marked in red in the paper. The main corrections in the paper and the responses to the reviewer’s comments are as following:
1. The mechanism diagram in Figure 2 still needs further refinement to distinguish whether the relevant action mechanisms involve transmembrane processes and whether they pertain to regulatory events within the cell nucleus. Additionally, the author should clearly differentiate and explain the upstream-downstream relationships of the signaling pathways in the Figure 2.
Answer: We sincerely appreciate your recognition of our work. As you suggested, we have refined the mechanistic diagram in Figure 2 to annotate nuclear regulatory events, enabling precise delineation of transmembrane transduction processes. Concurrently, modified figure legends explicitly distinguish and explicate signal relay between upstream and downstream pathway components. Revision has been made according to the comment and all the changes were marked in red color in the Marked-up Revised Manuscript (Page 3-4, figure 2, line 105-119).
We have tried our best to improve the manuscript and made changes in the manuscript. Meanwhile, we appreciate for your warm work earnestly, and hope that the correction will meet with approval.
Once again, thank you very much for your comments and suggestions.
Best wishes to you!